# Giant magneto-photoluminescence at ultralow field in organic microcrystal arrays for on-chip optical magnetometer

Hong Wang[1,2,6], Baipeng Yin[1,6], Junli Bai[2,3,6], Xiao Wei[3,4], Wenjin Huang[5], Qingda Chang[1,2], Hao Jia [1,2], Rui Chen[1,2], Yaxin Zhai [5], Yuchen Wu [2,3] ✉ & Chuang Zhang [1] ✉

Optical detection of magnetic field is appealing for integrated photonics; however, the light-matter interaction is usually weak at low field. Here we observe that the photoluminescence (PL) decreases by > 40% at 10 mT in rubrene microcrystals (RMCs) prepared by a capillary-bridge assembly method. The giant magneto-PL (MPL) relies on the singlet-triplet conversion involving triplet-triplet pairs, through the processes of singlet fission (SF) and triplet fusion (TF) during radiative decay. Importantly, the size of RMCs is critical for maximizing MPL as it influences on the photophysical processes of spin state conversion. The SF/TF process is quantified by measuring the prompt/delayed PL with time-resolved spectroscopies, which shows that the geminate SF/TF associated with triplet-triplet pairs are responsible for the giant MPL. Furthermore, the RMC-based magnetometer is constructed on an optical chip, which takes advantages of remarkable low-field sensitivity over a broad range of frequencies, representing a prototype of emerging opto-spintronic molecular devices.

Magneto-optical phenomenon, typically the Faraday/Kerr rotation of transmitted/reflected light polarization under an external magnetic field, has been discovered and extensively studied since 1840s[1,2]. After that, a series of magneto-optical effects, such as magnetic circular dichroism and magneto-chiral dichroism, have been found[3–9] and used for light modulation with magnetic field[10–14] and/or optical magnetometry[15–18]. A distinct type of magneto-optical effects concerning the excited state in organic materials, such as magneto-photoluminescence (MPL)[19–31] and magneto-electroluminescence[30–37], has attracted considerable attention in past decades. It refers to the change of luminescence intensity based on the spin conversion between singlet and triplet excited states, which is useful for both studying spin-related phenomena in existing optoelectronic devices[27–29,32–36] and exploring intriguing device architectures for opto-spintronics applications[14,15,38]. According to the spin-selection rule, the photoexcitation generates only singlet excitons that should be immune to an external magnetic field, and thus the magnitude of MPL is far below that of magneto-electroluminescence involving electrically injected triplet states[29–31]. The optical magnetometer via MPL is expected to be superior in integrated photonics, benefiting from its fast response and compact size[39–41]. Elevating the MPL performance is thereby essential and requires organic luminescent materials that can photo-generate triplet excitons from spin conversion processes[19–21].

Phosphorescent emitters (mainly heavy metal-organic complexes), in which triplet excitons are efficiently photo-generated through the singlet-to-triplet intersystem crossing due to strong spin-orbit

[1]Key Laboratory of Photochemistry, Beijing National Laboratory for Molecular Sciences, Institute of Chemistry, Chinese Academy of Sciences, Beijing, China. [2]University of Chinese Academy of Sciences, Beijing, China. [3]Key Laboratory of Bio-inspired Materials and Interfacial Science, Technical Institute of Physics and Chemistry, Chinese Academy of Sciences, Beijing, China. [4]Ji Hua Laboratory Foshan, Guangdong, China. [5]Key Laboratory of Low-Dimensional Quantum Structures and Quantum Control of Ministry of Education, Department of Physics, Hunan Normal University, Changsha, China. [6]These authors contributed equally: Hong Wang, Baipeng Yin, Junli Bai. ✉e-mail: wuyuchen@iccas.ac.cn; zhangc@iccas.ac.cn

coupling[20], remain insensitive to the magnetic field due to the lack of reversibility in spin conversion processes. Intermolecular charge transfer exciplexes can minimize the spin exchange energy between singlet and triplet, $\Delta E_{ST}$, and thus allow for both intersystem crossing and reverse intersystem crossing; thereby, the MPL magnitude increases to the level of ~5%[21,27,30], but is still far below the corresponding magneto-electroluminescence magnitude (>30%)[35–37]. In addition to the intersystem crossing and reverse intersystem crossing spin conversion, singlet fission (SF) and triplet fusion (TF) processes (between one singlet and two triplets) provide a promising way to realize the spin conversion by involving two adjacent molecules and consequently enhanced MPL[13–15,23–26,28,42–44]. For instance, acene derivatives involve both SF and TF processes and show a relatively large MPL of ~10% (close to the magnitude of magneto-electroluminescence at a ~200 mT field in previous reports[15,24]. We propose that the MPL performance could be significantly improved by facilitating the spin conversion of geminate SF and TF process, and a sufficiently large MPL at low fields under ambient conditions would hold great promise in the integrated magneto-optical devices such as on-chip optical magnetometer.

In this work, we report a giant MPL as high as >40% at a low magnetic field of ~10 mT in the high-quality rubrene microcrystals (RMCs) at room temperature, which is far above all the previously reported values for MPL in pi-conjugated molecules or polymers[15,19,24]. The capillary-bridge assembly preparation ensures the excellent crystallinity of RMCs, which is essential for the spin conversion processes of SF and TF, and moreover, it allows for precise control over the size of RMCs to maximize MPL. The unique size dependence of MPL is revealed and interpreted by the competition between SF and radiative recombination of singlets, as well as between geminate TF and dissociation of triplet-triplet pairs, studied by transient absorption and PL spectroscopies. It is further validated by the temperature-dependent PL, PL dynamics, and MPL, which correlate the thermally activated nature of the SF process in RMCs with the enhanced MPL at room temperature. The giant MPL, together with the facile fabrication of RMC arrays, enables the construction of an on-chip optical magnetometer that can detect an alternative field down to ~μT at a broad range of frequencies from ~Hz to ~MHz. These results offer a proof-of-concept for the spin manipulation of excited states and the exploration of magnetic field effects in organic materials towards novel opto-spintronic applications.

## Results

Rubrene is an acene derivative where four phenyl substituents attach to the planar tetracene backbone, and the energy of its first excited singlet state ($S_1$ ~ 2.23 eV) is very close to twice the energy of its lowest excited triplet state ($T_1$ ~ 1.14 eV)[45]. The small difference between $S_1$ and $2T_1$ (~ 50 meV) allows for the simultaneous occurrence of two spin conversion processes, i.e., SF and TF, in rubrene crystals[46], which enables the magnetic field effects on spin states of photo-excitations therein[15,24] (Supplementary Fig. 1). The SF process competes with direct recombination of $S_1$ and give rises to triplet pair state ($T_1$-$T_1$), which can participate in prompt PL via the geminate TF process or dissociate into separated $T_1$. Note that these photo-generated $T_1$ can recombine to $S_1$ via the bimolecular non-geminate TF process and give rise to delayed PL[47,48]. As shown in Fig. 1a, the spin conversion processes are valid only in solid state, and strongly related to the molecular packing of rubrene[49]. Compared to the thin films where the grain boundaries introduce additional radiative or non-radiative decay routes, the single crystals allow for more efficient spin conversion processes due to the highly ordered molecular packing. Here we adopted the capillary-bridge assembly method (Supplementary Fig. 2) to prepare high-crystalline rubrene microwire arrays, in which the dimensions of RMCs are precisely controlled by the dewetting dynamics of its solution[50]. The X-ray diffraction patterns in Fig. 1b indicate that the rubrene samples were crystalized in the triclinic phase. The polycrystalline film with an averaging grain size below

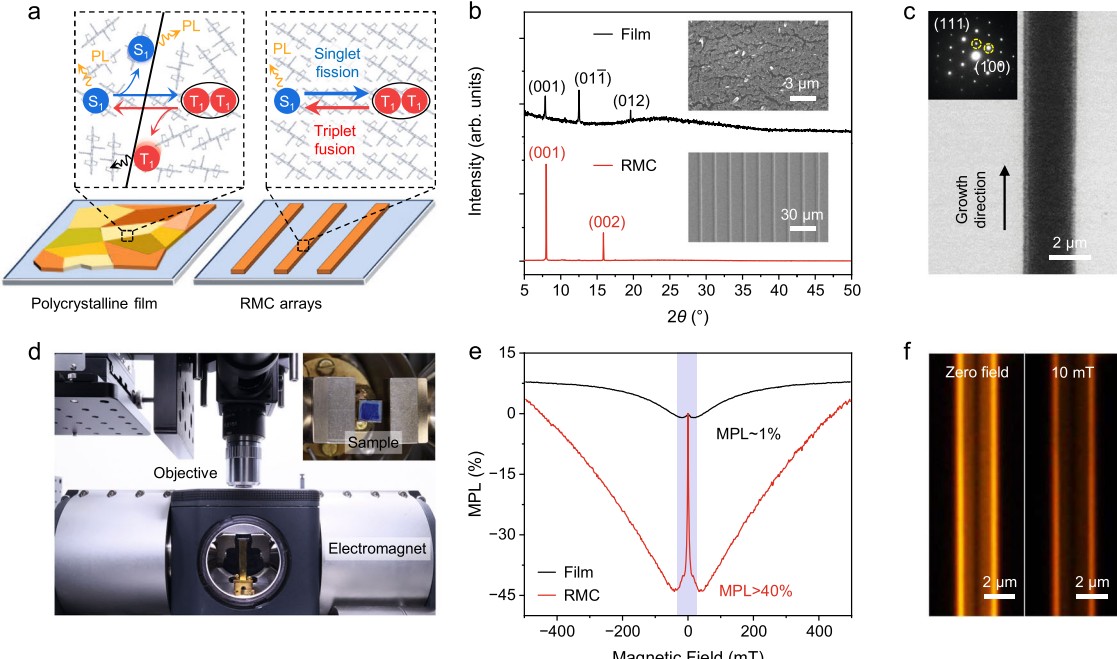

**Fig. 1 | Observation of giant MPL in RMCs. a** Illustration for the spin conversion processes in rubrene film and RMCs. **b** X-ray diffraction patterns of rubrene film (black) and RMCs (red). The corresponding scanning electron microscopy images are shown in the inset. **c** Transmission electron microscopy image and corresponding selected area electron diffraction pattern (inset) of a single RMC. **d** Photograph of the experimental apparatus for MPL measurements on RMCs.

**e** Typical MPL(**B**) curves measured from rubrene film (black) and RMCs (red) up to a field strength of 500 mT. The marked area indicates the MPL(**B**) curves at a low field <30 mT. **f** Optical images of a single RMC at zero field (left) and at 10 mT (right) under unfocused ultraviolet excitation. Source data are provided as a Source Data file.

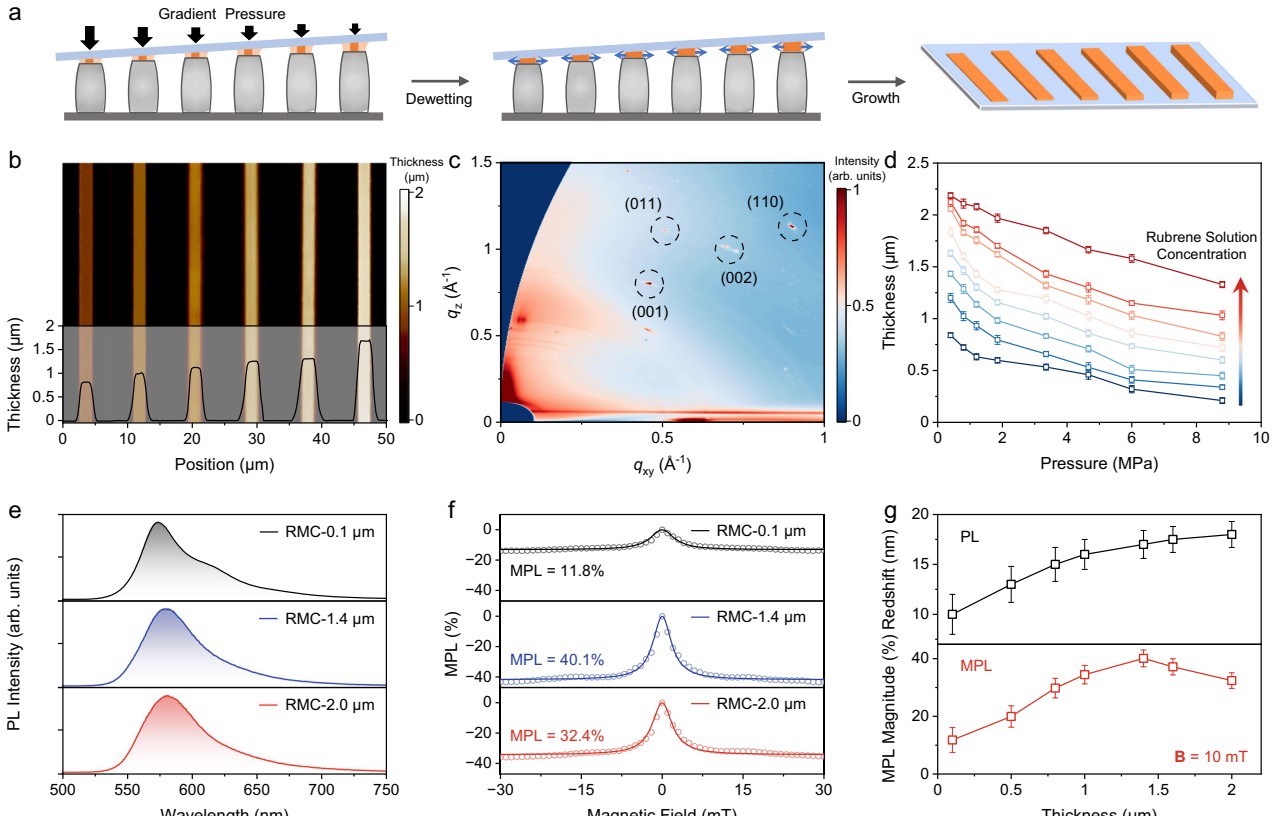

**Fig. 2 | Analysis on size-dependent MPL in RMCs. a** Illustration for the preparation of RMCs with various thicknesses under linear gradient pressures. **b** Atomic force microscopy image of RMCs on the same substrate varying from 0.8 to 1.7 μm in thickness. **c** Grazing-incidence wide-angle X-ray scattering pattern of the RMCs shown in (**b**). **d** Statistics on the thickness of as-prepared RMCs versus the applied pressure (from 0.5 to 9 MPa) using the rubrene solutions at 5, 8, 11, 14, 16, 18, 20, and 22 mg mL$^{-1}$ (from blue to red in sequence). Error bars indicate standard deviation from $n = 5$ independent replicates. **e** PL spectra measured from RMC-0.1 μm, RMC-1.4 μm, and RMC-2.0 μm. Excitation wavelength is 405 nm. **f** MPL(**B**) curves measured from RMC-0.1 μm, RMC-1.4 μm, and RMC-2.0 μm up to a field strength of 30 mT. **g** Plots of PL redshift (black) from polycrystalline film and MPL magnitude (red) at **B** = 10 mT as a function of the thickness of RMCs. Error bars indicate standard deviation from $n = 3$ independent replicates. Source data are provided as a Source Data file.

100 nm (Supplementary Fig. 3), shows (001), (01$\bar{1}$) and (012) X-ray diffraction peaks, corresponding to the three crystal planes exposed on the surface (Supplementary Fig. 4)[51]. In comparison, only the sets of (001) planes are exposed on the surface RMCs, which indicates the out-of-plane packing of rubrene molecules is perfectly oriented along its crystallographic *c* axis. The scanning electron microscopy images in the insets show that the large-area arrays of RMCs are well-aligned, of which the width is ~2 μm and the adjacent distance is ~15 μm. As shown in Fig. 1c, the transmission electron microscopy image and the selected area electron diffraction patterns are used to determine the in-plane molecular packing in RMCs. The facets in the short/long axis of RMCs consist of (111)/(100) planes, respectively. The highly ordered molecular packing in both out-of-plane and in-plane directions indicates the single-crystalline nature of as-prepared RMCs, and thus enables efficient SF and TF.

MPL measurements were carried out on a home-built optical microscope equipped with a magneto-optical cryostat (Fig. 1d). An in-plane magnetic field was applied to the RMC substrate placed in a vacuum chamber, and PL was collected by a non-magnetic objective and conducted to a spectrometer (Supplementary Fig. 5). The vacuum environment can prevent the quenching on PL/MPL due to the oxygen-sensitivity of triplets (Supplementary Fig. 6), and the degradation of RMC samples is negligible at the excitation power density of 100 mW mm$^{-2}$ (Supplementary Fig. 7), The MPL magnitude at a given magnetic field **B** is defined as MPL(**B**) = [PL(**B**)-PL(0)]/PL(0)×100% where PL(0) is the PL intensity measured at zero field, and the MPL curves are repeated for forward and backward scans (Supplementary Fig. 8).

As shown in Fig. 1e, the MPL(**B**) curve of rubrene film shows a W-like line shape and a low-field MPL component of ~1%, which is consistent with the results in previous reports[15,24]. Interestingly, the low-field MPL is significantly elevated in RMCs, which is tens of times larger than that of rubrene film and shows a dramatic decrease in PL intensity over 40% at a 10 mT field, far above the previously reported values (Supplementary Fig. 9). The unprecedented giant MPL is validated by the direct observation on PL images of RMCs under applied field, as shown in Fig. 1f. The bright PL is outcoupled from both edges of a single RMC due to the optical waveguiding in defect-free single crystals, and the >40% PL decrease under a 10 mT field is visualized to be uniform over the longitude of RMC. In comparison, the spatially resolved MPL in poly-crystalline rubrene film (Supplementary Fig. 10) shows a non-uniform distribution due to the existence of crystalline grains, which is consistence with the MPL enhancement in RMCs and also implies the dependence of MPL on crystal sizes.

As shown in Fig. 2a, a gradient pressure (0.5 to 9 MPa) was employed in the capillary-bridge assembly method to possess the asymmetric contact between the substrate and photoresist template with a small inclination angle[50]. Due to the difference in the volume of confined space, the nucleation-growth of rubrene during the continuous dewetting of its solution yields the thickness tailoring of RMCs on the same substrate. The atomic force microscopy image in Fig. 2b indicates that the thickness of RMCs is precisely controlled and gradually increases from the high-pressure side to the low-pressure side. All the RMCs on the same substrate are perfectly orientated, proved by the sharp diffraction spots in grazing-incidence wide-angle X-ray

scattering patterns (Fig. 2c), and the single-crystalline nature and molecular packing of RMCs remain unchanged although their thickness varies (Supplementary Fig. 11 and Supplementary Fig. 12). The thickness of RMCs is further tuned from ~0.1 μm to ~2 μm, by using the precursor solutions of 5 to 22 mg mL$^{-1}$ in rubrene concentration, as shown in Fig. 2d. Note that the lateral dimensions of RMCs would not influence the MPL effect obviously (Supplementary Fig. 13 and Supplementary Fig. 14). The tailoring of RMC thickness allows for the investigation on the size-dependent MPL based on the excited state processes because the spin conversion processes of SF and TF are found to be influenced by the crystal sizes in rubrene[52–54].

The PL band of monomer rubrene in its dilute solution (green line in Supplementary Fig. 15) is centered at ~560 nm followed by a vibronic sideband at ~600 nm[55]. In comparison, the PL of rubrene film (blackline in Supplementary Fig. 15) contains both the monomer feature associated with the boundary/surface of crystal grains and the PL contribution from bulk state that undergoes the spin conversion processes of SF and TF. As shown in Fig. 2e, the PL band of RMC-0.1 μm is significantly red-shifted compared to that of polycrystalline film, which implies that the spin conversion processes are facilitated in the single crystal of rubrene. The vibronic feature is still noticeable in RMC-0.1 μm, meaning that the singlet excitons prefer to recombine directly in the monomer rubrene. As the thickness of RMCs increases, the monomer feature in PL disappears, and the PL band red-shifts further to ~580 nm. In this case, the spin conversion processes of geminate SF and TF dominate the PL and thus enable the observation of large MPL upon spin conversion, which is validated by the absence of MPL (< 0.1%) from the rubrene solution (Supplementary Fig. 16). As shown in Fig. 2f, the MPL in RMC-0.1 μm is ~11.8% at 10 mT, while the MPL in RMC-1.4 μm follows the same line shape but its magnitude reaches ~40.1%. The MPL magnitude slightly decreases at a high excitation power density in RMCs (Supplementary Fig. 17), while it remains at a very low level in polycrystalline films (Supplementary Fig. 18). It indicates that the 1.4 μm thickness promotes the spin conversion through geminate SF and TF processes in highly ordered structures. Surprisingly, the MPL magnitude in RMC-2.0 μm decreases to 32.4%, although its PL is almost the same as that in RMC-1.4 μm. The intriguing size dependence of PL red-shift (Supplementary Fig. 19) and MPL magnitude is summarized in Fig. 2g, which indicates that the MPL magnitude increases with the red-shift of PL and starts to decrease when the red-shift is saturated. The dependence agrees with the further reduced MPL magnitude of ~17% in the bulk crystal of rubrene (Supplementary Fig. 20) which should also be related to the interplay between SF and TF processes. The size dependence on giant MPL in RMCs is verified by evaluating other possible contributions from molecular packing and crystal anisotropy, through the MPL results in various rubrene crystal phases (Supplementary Fig. 21), the angle-resolved MPL measurements on bulk crystals (Supplementary Fig. 22) and RMCs (Supplementary Fig. 23)[42–44].

The temporal evolution of SF and TF processes, spanning the femto- to nano-second time scales, is examined through ultrafast transient absorption spectroscopy. As shown in Fig. 3a and Supplementary Fig. 24, the time-dependent transient absorption spectra in both RMC and thin film samples have revealed a complex landscape of absorption features. The short-lived feature centered at 440 nm corresponds to the characteristic singlet absorption ($S_1 \rightarrow S_n$), while the longer-lived feature at 510 nm is indicative of typical triplet absorption ($T_1 \rightarrow T_n$)[56]. The feature at 480 nm could be possibly related to the absorption of ($T_1$-$T_1$) due to its similar trend to $T_1$. The emergence of the triplet state within 1 ps is a compelling confirmation of the SF process in both samples, considering that the conventional intersystem crossing typically happens in tens of nanoseconds. It is noteworthy that the RMC sample exhibits significantly prolonged lifetimes for both singlet and triplet states in comparison to the thin film counterpart (Supplementary Fig. 25). This distinction becomes

particularly pronounced after the 200 ps, where photo-excitation attains a plateau extending up to 5 ns (the limit of our transient absorption setup). This prolonged equilibrium state between geminate SF and TF processes grants us ample time to manipulate the singlet emission through a magnetic field in Fig. 1e. To further elucidate this intriguing state in RMCs, we extend our observations up to 900 ns via transient PL, as depicted in Fig. 3b. Compared to the PL decay for monomer rubrene in dilute solution, the PL decays in RMCs are shortened in the prompt regime (<100 ns) due to the geminate SF and TF processes, and show the non-geminate-TF-induced delayed components in the time regime of >100 ns. The average lifetime of prompt PL $\tau_{prompt}$ is used to evaluate the amount of remaining singlets that do not undergo SF process, and the power function factor of delayed PL $m_{delayed}$ is used to profile the amount of free triplets generated from the dissociation of ($T_1$-$T_1$), which participate in the delayed PL through non-geminate TF. It is observed that the quantity of singlet exciton that undergoes the SF process increases (the blue area) while the dynamics of non-geminate-TF-induced delayed PL becomes slower (the red area) with the increased thickness from RMC-0.1 μm to RMC-2.0 μm.

The size dependence of SF and TF processes (Supplementary Fig. 26 and Supplementary Table 1) is analyzed in Fig. 3c, which should be responsible for the size-dependent MPL as well as the giant MPL observed in RMCs[52–54]. The size-dependent photophysical properties are probably due to the cooperative singlet and triplet exciton transport[57], which could occur on the same scale of RMC thickness especially considering the penetration depth of excitation light. In RMCs, the competition of photophysical processes in singlets (direct recombination and SF) and in triplet-triplet pairs (geminate TF and dissociation into triplets) determines the MPL magnitude (up to 40%). (i) For RMC-0.1 μm, the direct radiative recombination of $S_1$ is more efficient than the SF process into ($T_1$-$T_1$) for spin state conversion; (ii) For RMC-1.4 μm, SF process is facilitated and allows for the reversible conversion between $S_1$ and ($T_1$-$T_1$) through geminate SF and TF processes; (iii) For RMC-2.0 μm, the dissociation of ($T_1$-$T_1$) into $T_1$ is more pronounced and diminishes the spin state conversion; in addition, $T_1$ participates in non-geminate TF that brings an opposite MPL effect. It is confirmed by the smaller $m_{delayed}$ obtained from the analysis of delayed PL in bulk rubrene crystals (Supplementary Fig. 27). Moreover, the transient MPL is obtained by measuring and comparing the PL decays at 0 mT and 10 mT, as shown in Fig. 3d. The MPL response appears immediately after photoexcitation and reaches the maximum value at a time delay of ~2 ns, indicating that the giant negative MPL effect should be attributed to the rapid geminate SF and TF processes. As a result, the spin conversion routes based on SF and TF processes are optimal in (ii) RMC-1.4 μm where the geminate SF and TF are promoted over the direct recombination of $S_1$ and the dissociation into $T_1$, respectively, which explains its superior low-field MPL performance >40%.

Our interpretation is further proven by the PL and MPL results measured at cryogenic temperatures, as the SF process in rubrene should be de-activated when the thermal energy k$T$ is far below the energy difference $\Delta E$ between $S_1$ and $2T_1$ ( ~50 meV)[58]. In this case, the monomer PL becomes obvious, and the spin conversion is thus hindered. The PL spectra of RMC-1.4 μm exhibit more vibronic features at low temperature that could be assigned to the monomer PL of rubrene (Supplementary Fig. 28). Meanwhile, the PL intensity increases dramatically when cooling down the RMCs, as the prompt PL from geminate TF in RMCs at low temperature is much less efficient than direct recombination in dispersed rubrene in polymer matrix (Supplementary Fig. 29). The lifetime of monomer PL, $\tau_{monomer}$, from dispersed rubrene shows a very weak dependence on temperature (Supplementary Fig. 30). Thus it is still reasonable to use $\tau_{prompt}$ to describe the contribution of SF process at different temperatures (Supplementary Fig. 31 and Supplementary Table 2). As shown in Fig. 3e, the temperature dependence of $\tau_{prompt}$ from transient PL decay

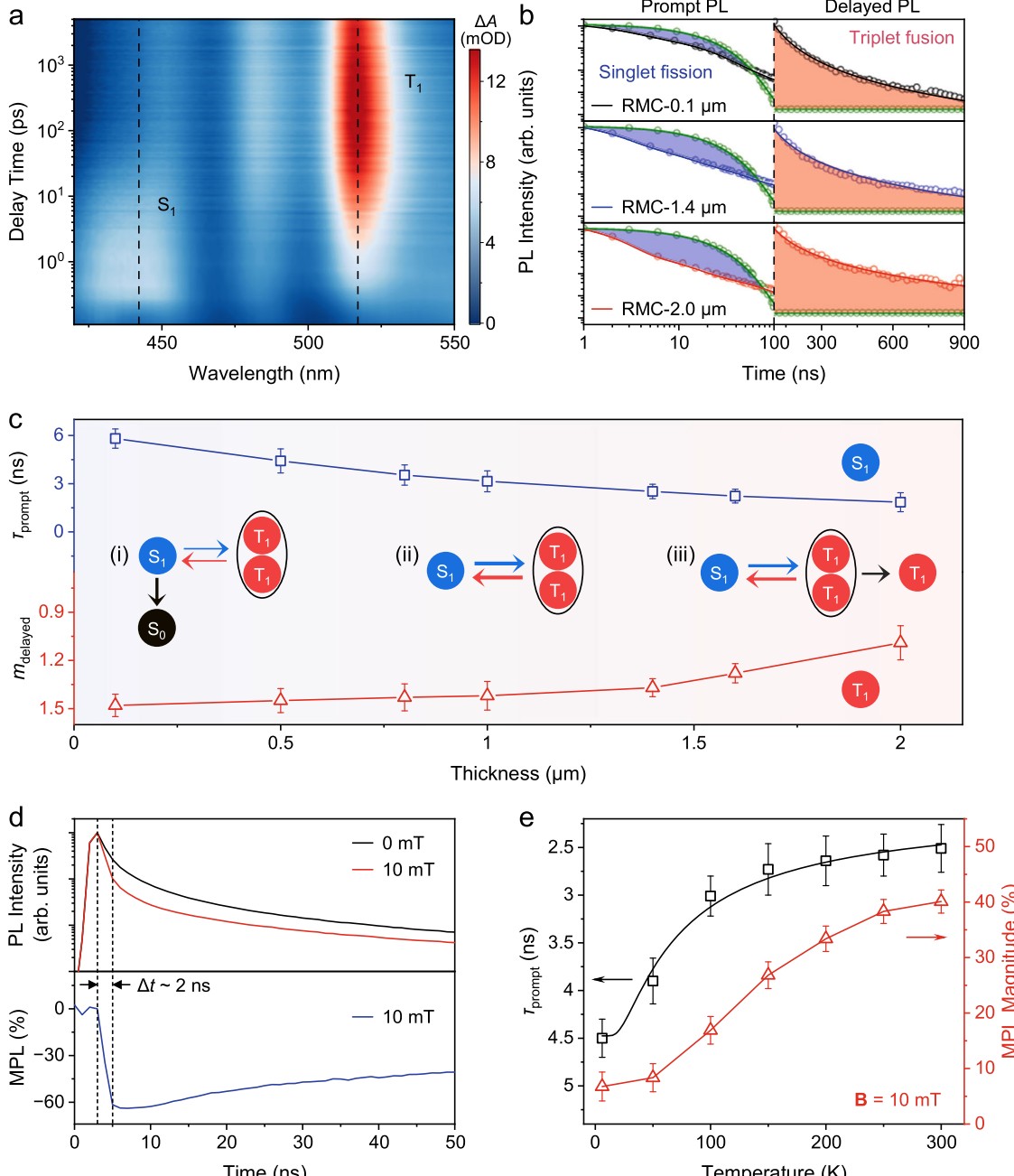

**Fig. 3 | Singlet-triplet spin conversion through SF and TF in RMCs. a** Pseudo-color image of time-dependence transient absorption spectra in RMC. $\Delta A$ is the change of absorption intensity. The excitation wavelength is 400 nm. **b** Transient PL decay curves measured from RMC-0.1 μm, RMC-1.4 μm, and RMC-2.0 μm. Transient PL decay measured from rubrene solution (green) is also included. The PL decays are decomposed into the prompt (<100 ns) and delayed (>100 ns) regimes. Note that the curves in the delayed regime (>100 ns) are re-normalized for a better comparison. The blue and red areas indicate the decreased prompt PL due to singlet fission and the increased delayed PL due to non-geminate triplet fusion,

respectively. **c** The lifetime of prompt PL ($\tau_{prompt}$, blue) and the power function factor of delayed PL ($m_{delayed}$, red) as a function of the thickness of RMCs. Illustration for the size effect on SF and TF processes in RMCs is shown in the inset. Error bars indicate standard deviation from $n = 3$ independent replicates. **d** PL decay curves of the RMC-1.4 μm measured at 0 mT and 10 mT (top), and the corresponding transient MPL response at 10 mT (bottom). **e** Averaging lifetime of prompt PL (black) and MPL magnitude (red, at 10 mT) as a function of temperature for RMC-1.4 μm. Error bars indicate standard deviation from $n = 3$ independent replicates. Source data are provided as a Source Data file.

curves shows an onset point of ~50 K and is saturated near room temperature, which can be well fitted by the thermal distribution function based on the Arrhenius equation. Notably, the temperature-dependent MPL magnitude (Supplementary Fig. 32) follows the same trend, indicating that the MPL effect is semi-quantitatively associated with the geminate SF and TF processes in RMCs. Therefore, the promoted geminate SF and TF processes are essential to the observed giant MPL in RMCs.

The giant magneto-optical effect at low field allows for the construction of an integrated optical magnetometer based on RMC at room temperature. As shown in Fig. 4a, RMCs are fabricated on the pre-patterned polymethyl methacrylate (PMMA) waveguides, and the light signals are coupled from/to the network of optical waveguides on the photonic chip. The efficient near-field coupling between the RMC and the PMMA waveguide ensures the detection of the low magnetic field at different frequencies generated by a Helmholtz coil[59]. In the

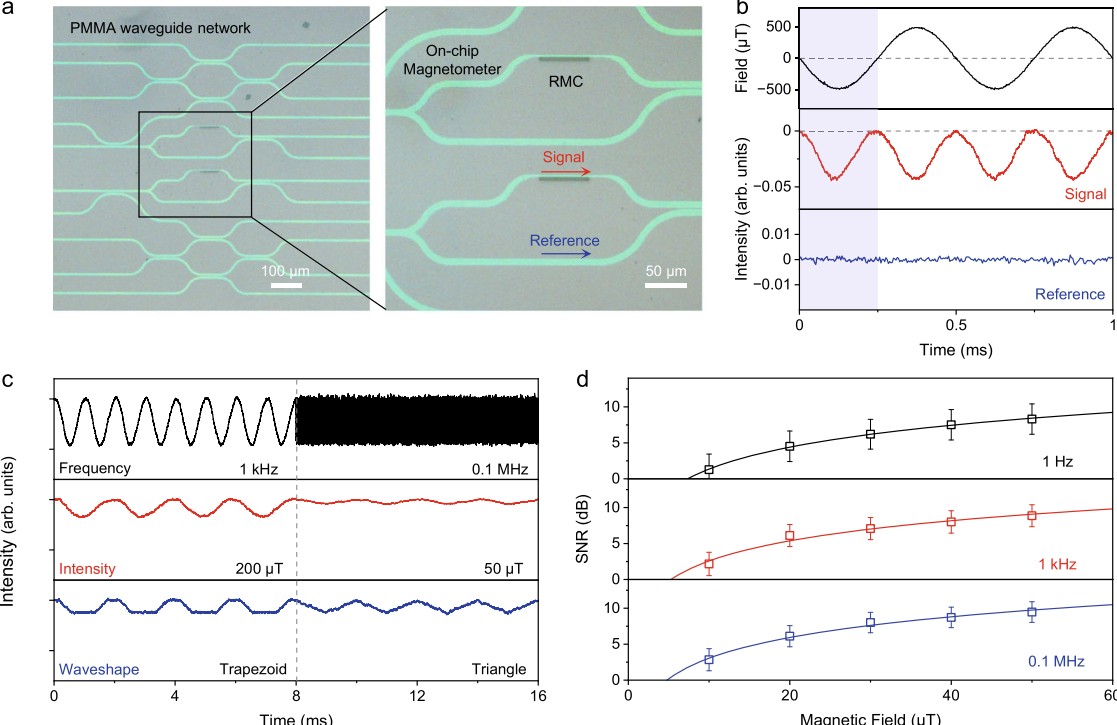

**Fig. 4 | RMC-based on-chip optical magnetometer. a** Optical microscopy image of the on-chip RMC magnetometer and the zoom-in image on the on-chip RMC magnetometer. **b** Response of RMC magnetometer in signal (red) and reference (blue) under applied alternative field (black). The shaded region indicates the signal and reference under half the period of the applying field. **c** The detected signal response from the on-chip RMC magnetometer at different frequency (black), intensity (red) and waveshape (blue) of alternative field. **d** Signal-to-noise ratio (SNR) values of RMC magnetometer obtained under applied magnetic field at different frequencies of 1 Hz (black), 1 kHz (red) and 0.1 MHz (blue). Error bars indicate standard deviation from $n = 3$ independent replicates. Source data are provided as a Source Data file.

RMC-based on-chip magnetometer, the light passing RMC is considered as the signal, while the light through the bypass waveguide is used as the reference for zero-field intensity. In principle, the nanosecond response of MPL in RMCs offers possibilities for magnetic field detection up to ~GHz frequency. As shown in Fig. 4b, a cosine-modulated light signal at 2 kHz is outcoupled from the PMMA network when the alternative field between -500 μT and +500 μT is applied at 1 kHz. Meanwhile, the reference is constant under the external field and used to determine the noise level of the optical magnetometer. As shown in Fig. 4c, the external field is detected without any obvious attenuation of signal amplitude in a broad range of frequencies ramping up to 0.1 MHz, which is actually limited by the highest frequency in operating the Helmholtz coil. Moreover, the RMC magnetometer can detect a low field down to ~μT level benefiting from the giant MPL effect, and the waveshapes of alternative fields (e.g., trapezoid, triangle) can even be accurately recognized. The signal-to-noise ratio (SNR), $SNR = 10 \times lg(MPL_{signal}/NOISE_{reference})$, is adopted to profile the sensitivity of on-chip optical magnetometer, as shown in Fig. 4d. The SNR values follow a power-law function of the field strength in a broad range of frequencies from Hz to MHz, and the intercept represents the limit of detection below 10 μT. These parameters of RMC devices are comparable to state-of-art integrated optical magnetometers[17,18], showing the advantages of unique fabrication techniques and magneto-photophysical properties from organic semiconductors.

## Discussion

In summary, we have achieved a giant MPL of over 40% at a 10 mT field at room temperature in RMCs fabricated by the capillary-bridge assembly method. The remarkable magnetic field effect relies on the reversible spin conversion of geminate SF and TF between $S_1$ and $(T_1-T_1)$, which can be finely controlled by the size effect on photophysical processes in the low-dimensional RMCs. The roles of photogenerated $S_1$ and $(T_1-T_1)$ in MPL are revealed by the steady state and transient spectroscopies. A proof-of-concept for an on-chip optical magnetometer is demonstrated based on RMCs, and it can detect a local magnetic field at ultralow intensity (~μT) and broad frequency (~Hz to MHz). Our work offers a promising way to enhance magneto-optical effects in organic semiconductors and illustrates their potential applications in active-response integrated photonic devices.

## Methods
### Fabrication
Rubrene was purchased from Innochem and used without further purification for the preparation of RMCs. The rubrene powders were dissolved in analytically pure chloroform to prepare the precursor solution. The photoresist template with micropillars was employed for guiding the dewetting of rubrene solution. 6 μL rubrene solution was dropped onto the template and was covered by a silicon substrate. The systems were dried at room temperature in a vacuum oven. The polycrystalline rubrene was prepared by the drop casting method. The bulk single crystal was grown analytically in 1-propanol without additional processing from the saturated solution.

### Characterizations
The optical image of the RMC magnetometer was acquired by an optical microscope (LV-IM, Nikon) integrated with a Helmholtz coil (Hunan Paisheng Technology Ltd), which can generate an alternating magnetic field. The morphologies of polycrystalline rubrene and RMC arrays were investigated by scanning electron microscopy (JSM-7500F, JEOL). The transmission electron microscopy image and selected area

electron diffraction patterns were characterized on JEM-2100F (JEOL) at an accelerating voltage of 200 kV. The atomic force microscopy measurements of RMCs were acquired on MultiMode 8 (Bruker). The X-ray diffraction diagrams of polycrystalline rubrene and RMC arrays were measured on D8 diffractometer (Bruker). Grazing-incidence wide-angle X-ray scattering is performed at the Beijing Synchrotron Radiation Facility (BSRF) at a 0.2° incidence angle. The thickness of RMCs for the statistics was measured on the confocal laser scanning microscope (OLS-4500, Olympus).

## Optical and magneto-optical measurements

All optical and magneto-optical measurements were carried out in the vacuum chamber with a pressure <2.0 torr in a cryostat integrated with electromagnet (Cryostation s50, Montana Instruments). Steady-state PL measurements including temperature-dependent and thickness-dependent PL spectra and MPL were performed with 405 nm continuous-wave laser (DL5146-101S and LTC56A/M, Thorlabs) in the home-built optical characterization system (Supplementary Fig. 5). The spectra were acquired by a deep cooling CCD (PIXIS: 265E, Teledyne Princeton Instruments) connected to the spectrograph (SpectraPro HRS-300-MS, Teledyne Princeton Instruments). MPL signal was acquired by a silicon photodiode (SM1PD1A, Thorlabs) and collected after an amplifier (PDA200C, Thorlabs) by a digital multimeter (DMM6500, Keithley). All PL decays were performed with the time-correlated single photon counting (TCSPC) method with the filtered picosecond pulsed light source centered at 430 nm with FWHM of ~5 nm from a supercontinuum source (SC-PRO, YSL Photonics) and the PL signal was acquired by a single-photon counting module (SPCM-AQRH-14, Excelitas) and processed by a time-to-digital converter (quTAG, qutools). The output signal of RMC magnetometer under alternative field was measured by the same silicon photodiode system above at low frequency and by the photomultiplier tube (H10721-01, Hamamatsu) at high frequency.

## Transient absorption measurements

The transient absorption measurements were carried out using a pump-probe spectrometer. A Ti: sapphire amplifier generates the fundamental laser pulse with a wavelength of 800 nm and a pulse repetition rate of 1 kHz. This fundamental pulse is divided into two parts by a beam splitter. One part is directed to an optical parametric amplifier (OPA) to produce a 400 nm pump, which is then chopped at 500 Hz and attenuated using neutral-density filter wheels. The average excitation density is determined by dividing the input photon flux by the pump penetration depth. The other part of the fundamental pulse is focused into a sapphire crystal to create a white-light continuum (450-800 nm) that serves as the probe. A motorized translation stage with a retroreflecting mirror is used to delay the probe pulses in time relative to the pump pulses. The pump and probe beams are spatially overlapped at the sample's surface and incident normally. The focused spot size for the probe and pump beams at the sample position is ~200 μm and 600 μm, respectively.

## Data availability

All data generated in this study are provided in the Supplementary Information and the Source Data file. Additional raw data are available from the corresponding authors upon request. Source data are provided in this paper.

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

## Acknowledgements

This work was financially supported by the Ministry of Science and Technology of China (Nos. 2018YFA0704802 C.Z., 2018YFA0704803 Y.W., 2020YFA0714603 C.Z.), the National Natural Science Foundation of China (Nos. 12204167 Y.Z., 22090021 C.Z., 52173190 Y.W., 21988102 Y.W.), the CAS "Strategic Priority Research Program B" (No. XDB0520302 C.Z.), and the Youth Innovation Promotion Association CAS (No. 2018034 Y.W.). We acknowledge Prof. Chang-Ling Zou at the University of Science and Technology of China, Prof. Li-Zhen Huang at Soochow University, and Prof. Chuan-Xiang Sheng at Fudan University for the fruitful discussion. A portion of this work is based on the data obtained at BSRF-1W1A. The authors gratefully acknowledge the cooperation of Dr. Yu Chen at the BSRF-1W1A beamline.

## Author contributions

Y.Z., Y.W., and C.Z. supervised the project. H.W., B.Y., J.B., X.W., Q.C., and R.C. fabricated the rubrene samples. J.B. and X.W. prepared the on-chip RMC magnetometer. B.Y., J.B., X.W., Q.C., and H.J. carried out the morphological and crystallographic characterization measurements. H.W. carried out the optical and magneto-optical measurements. W.H. carried out the transient absorption measurements. H.W. and B.Y. analyzed the data and wrote the draft. All authors discussed and contributed to the manuscript preparation.

## Competing interests

The authors declare no competing interests.
