## [Peer Review File · Nature Communications]

Giant magneto-photoluminescence at ultralow field in organic microcrystal arrays for on-chip optical magnetometerReviewer #1 (Remarks to the Author):

In this manuscript, Wang et al. report the observation of ultralarge magneto-photoluminescence in rubrene microcrystals. The underlying mechanism of singlet fission and triplet fusion is systematically investigated, and its potential application as magneto-optical modulators is also demonstrated. To me, the experimental result of >40% MPL at low field strength is striking, which is far above the values in previous studies. Their interpretation is convincing which is supported by the transient spectroscopies. Overall, I believe this work is of high quality, and could be suitable for publishing in Nature Communications, after the following issues being carefully addressed by the authors in a revised version.

1. As shown in Figure 1b, the rubrene samples used in this work are all in the triclinic phase. It is known that rubrene has several crystal phases (monoclinic, triclinic, and orthorhombic), and the molecular packing is quite different in these phases. The authors should consider the influence of crystal phases on MPL in rubrene.

2. The size-dependence of MPL is studied by controlling the thickness of rubrene crystals, and the results are interpreted by the confinement of exciton diffusion. How about the lateral dimensions of crystals? Is the width or length related to exciton diffusion and MPL?

3. The rubrene crystals, especially the single crystals used in this work, are anisotropic. I am curious whether the MPL is anisotropic. Can the authors change the field direction, or rotate the sample, when measuring MPL?

4. The PL intensity of organic luminescent materials usually decreases with time due to the sample degradation under optical excitation. How is the stability of PL from as-prepared rubrene crystals? In addition, the authors should provide a forward scan and a backward scan of MPL to exclude the influence of sample degradation.

5. What about the MPL of rubrene crystals measured in ambient condition? I assume the photogenerated triplets are sensitive to oxygen, so the MPL in air should be different from that in vacuum.

6. The authors should examine the influence of excitation power on MPL, because the process of triplet fusion is known to be power-dependent.

7. The concept of magneto-optical device shown in Figure 4 is interesting. However, I am a bit confused by the device configuration. Why the rubrene crystal needs to be so long? The active area should be between the two optical fibers, so the crystal on two sides would introduce additional loss due to optical waveguiding.

8. The >40% MPL is a remarkable value as far as I know. I recommend the authors to provide a summary of MPL values in previous studies for comparison. It would more clearly show the significance of this work in terms of very large magnetic-optical effects.

Reviewer #2 (Remarks to the Author):

See the attached file.

Reviewer #3 (Remarks to the Author):

In the present manuscript, the authors prepared rubrene microcrystals using a capillary-bridge assembly method. Magnetic field effects on luminescence from the rubrene microcrystals were observed with a home-built optical microscope equipped with a magneto-optical cryostat. The decrease of fluorescence by 40 % was observed at 100 mT. Fluorescence intensity was modulated by oscillating external magnetic field using Helmholtz coil.

Although physical phenomena observed in the present manuscript may be interesting, the general readers of nature communications will not understand or be interested in this topic. I do not recommend publication of the present manuscript in Nature Communications for the following reasons.

1.The authors claimed that the present material can be a prototype of emerging opto-spintronic devices. However, I can't imagine exactly how this material would be useful for device development. In introduction, the authors also mention about magneto-photoluminescence with several literatures. In these literatures, however, it is not explained how magneto-photoluminescence improves existing devices or contributes to develop new functional devices. In addition to that, I do not think that luminescence from the rubrene crystal is coherent, which is important in development of quantum optical devices.

2.The authors claimed that the observed large magnetic field effects were due to the confinement of triplet excitons in micro-space. In our laboratory, we also measured magnetic field effects on fluorescence from rubrene crystals with a variation of crystal phase. The figure attached shows an example of our data. Although the crystals are visible, we can observe the 25 % decrease at 50 mT. Therefore I can not believe that the magnetic field effects in the present manuscript is purely caused by the confinement of excitons. The authors changed the size of the materials by applying pressure. The pressure also changes the packing of crystals. It is not clear whether magneto-photoluminescence is affected by packing change or size change of the crystals.

3.The modulation of the fluorescence by the oscillating magnetic field has been used to study low magnetic field effects on radical pairs (C. T. Rodgers, Pure Appl Chem. 2009, 81, 19-43). Therefore, the technique in the manuscript is not new. I can not imagine how it works in organic devices. What means a magneto-optical integrated device?

4.In page 9, Discussion should be Summary.

Response to Reviewers' comments

Response to Reviewer #1

In this manuscript, Wang et al. report the observation of ultralarge magneto-photoluminescence in rubrene microcrystals. The underlying mechanism of singlet fission and triplet fusion is systematically investigated, and its potential application as magneto-optical modulators is also demonstrated. To me, the experimental result of >40% MPL at low field strength is striking, which is far above the values in previous studies. Their interpretation is convincing which is supported by the transient spectroscopies. Overall, I believe this work is of high quality, and could be suitable for publishing in Nature Communications, after the following issues being carefully addressed by the authors in a revised version.

Our Response: We appreciate Reviewer #1 for the overall positive evaluation and his/her time and efforts in examining our work. In the following, we have provided detailed responses to Reviewer #1's comments, and the main text and Supplementary Information have been revised accordingly.

1. As shown in Figure 1b, the rubrene samples used in this work are all in the triclinic phase. It is known that rubrene has several crystal phases (monoclinic, triclinic, and orthorhombic), and the molecular packing is quite different in these phases. The authors should consider the influence of crystal phases on MPL in rubrene.

Our Response: Reviewer #1 is correct that MPL in the monoclinic, triclinic, and orthorhombic phases of rubrene crystals should be different, as the molecular packing is crucial to the SF and TF processes [e.g., Wang, X. P. et al. *CrystEngComm*, **18**, 7353-7362 (2016)], and consequently determines the MPL effect.

In the revised version, we have supplemented the MPL measurements on the rubrene crystals in monoclinic, triclinic, and orthorhombic phases, as shown in **Fig. R1**.

The MPL(B) curves of three rubrene crystals show the same line shape, but their MPL magnitudes vary from $\sim 5\%$ (monoclinic), $\sim 12\%$ (orthorhombic), to $\sim 17\%$ (triclinic) at 10 mT. The large MPL in triclinic phase could be attributed to the highly-ordered slipped-parallel packing of rubrene molecules [e.g., Liu, Y. P. *et al. Aggregate* **4**, e347 (2023)].

Fig. R1. (a) XRD patterns of rubrene samples in monoclinic, orthorhombic, and triclinic phases. Insets show corresponding molecular packing styles. (b) MPL(B) curves measured from monoclinic, orthorhombic, and triclinic rubrene up to a field strength of 30 mT.

Fig. R2. XRD patterns measured from triclinic rubrene crystals and RMCs. The calculated XRD of rubrene in triclinic phase is also included.

It should be noted that our RMCs are triclinic according to the XRD results (**Fig. R2**), which shows the largest MPL in the three bulk crystals. The size effect of MPL was found in RMCs, leading to a giant MPL of $\sim 40\%$ by tuning thickness of RMCs while the same molecular packing was maintained in RMCs.

Accordingly, **Fig. R1** have added as Supplementary **Fig. 21** in Supplementary Information. The above discussion has been added in revised main text (page 6) and in Supplementary Information.

2. The size-dependence of MPL is studied by controlling the thickness of rubrene crystals, and the results are interpreted by the confinement of exciton diffusion. How about the lateral dimensions of crystals? Is the width or length related to exciton diffusion and MPL?

Our Response: According to Reviewer #1's comments, we have carried out the MPL measurements on RMCs with various lengths (**Fig. R3**) or widths (**Fig. R4**). The lateral dimensions of RMCs (with a constant thickness of 1.4 μm) are tuned from 20 to 60 μm in length, and from 2 to 8 μm in width. These samples show very similar MPL responses of $\sim 40\%$ at 10 mT. Therefore, we conclude that the MPL effect is not obviously influenced by the lateral dimensions of RMCs in our experiments. Note that the lateral dimensions (template-controlled) are significantly larger than the thickness (pressure-controlled) of RMCs due to the interface-confined assembly of our capillary-bridge method. In addition, the MPL effect should be more sensitive to the thickness of RMCs, since the excitation light is illuminated on the top surface and leads to exciton diffusion in the vertical direction.

Fig. R3. (a) Fluorescence images of RMCs with different lengths. Scale bar is 5 μm . (b) MPL(B) curves measured from RMCs with different lengths (20, 25 and 60 μm).

Fig. R4. (a) Fluorescence images of RMCs with different widths. Scale bar is 5 μm . (b) MPL(B) curves measured from RMCs with different widths (2, 4 and 8 μm).

Accordingly, **Fig. R3** and **Fig. R4** have been added as **Supplementary Fig. 13** and **Supplementary Fig. 14** in Supplementary Information, respectively. The above discussion has also been mentioned in the revised main text (page 5) and in Supplementary Information.

3. *The rubrene crystals, especially the single crystals used in this work, are anisotropic. I am curious whether the MPL is anisotropic. Can the authors change the field direction, or rotate the sample, when measuring MPL?*

Our Response: We agree with Reviewer #1 that the MPL effects should be anisotropic in rubrene crystals, due to the anisotropy in SF and TF processes in the crystal structure. Anisotropic MPL has been previously reported in anthracene crystals [**Ref. 42**, Johnson, R. C. *et al. Phys. Rev. B* **1**, 896-902 (1970)] and tetracene crystals [**Ref. 43**, Merrifield, R. E. *et al. Chem. Phys. Lett.* **3**, 386-388 (1969)], but not yet in rubrene crystals to the best of our knowledge. Since the crystal structure of rubrene is less anisotropic, the MPL is expected to be less anisotropic compared to that in tetracene crystals.

In the revised version, we have supplemented the MPL measurements on RMCs by changing the in-plane angle θ of external field relative to the long axis of RMCs, as shown in **Fig. R5**. The long axis is along the (111) direction of triclinic phase rubrene. The anisotropy of MPL is observed but relatively weak compared to the large MPL

magnitude ($\sim 40\%$). The maximum MPL appears at 40° and 130° , and the change of MPL magnitudes up on the direction of external field is $\sim 3\%$.

Fig. R5. (a) Fluorescence image of single RMC-1.4 μm . Scale bar is 5 μm . The definition of rotation angle θ according to the long axis along (111) of RMC. (b) Corresponding MPL magnitude obtained at 10 mT as a function of the rotation angle of magnetic field.

Accordingly, **Fig. R5** is added as **Supplementary Fig. 23** in Supplementary Information. The above discussion and citations (**Ref. 42** and **Ref. 43**) have also been mentioned in the revised main text (page 6) as well as in Supplementary Information.

4. The PL intensity of organic luminescent materials usually decreases with time due to the sample degradation under optical excitation. How is the stability of PL from as-prepared rubrene crystals? In addition, the authors should provide a forward scan and a backward scan of MPL to exclude the influence of sample degradation.

Our Response: According to Reviewer #1's comments, we have recorded the PL intensity from RMCs under a continuous excitation at the power density of 10, 100, 1000 mW/mm^2 , as shown in **Fig. R6**. In our MPL measurements, the typical excitation power is $100 \text{ mW}/\text{mm}^2$ and it usually takes less than 300 seconds to obtain a typical $\text{MPL}(B)$ curve. We have also provided a forward scan and a backward scan of MPL on RMCs, as shown in **Fig. R7**. The forward and backward scans are almost identical without any obvious hysteresis. Therefore, the sample degradation should be negligible in the contributions to MPL effects in our experiments.

Fig. R6. PL intensity measured from RMCs under different excitation power density of 10, 100, 1000 mW/mm².

Fig. R7. The forward (-500 mT to +500 mT) scan and backward (+500 mT to -500 mT) scan of MPL(*B*) curves under the excitation power density of 100 mW/mm².

Accordingly, **Fig. R6** and **Fig. R7** have been added as **Supplementary Fig. 7** and **Supplementary Fig. 8** in **Supplementary Information**, respectively. The above discussion has also been mentioned in the revised main text (page 4) and in **Supplementary Information**.

5. *What about the MPL of rubrene crystals measured in ambient condition? I assume the photogenerated triplets are sensitive to oxygen, so the MPL in air should be different from that in vacuum.*

Our Response: According to Reviewer #1's comments, we have carried out the MPL measurements on the same RMC in vacuum and in ambient condition, as shown in **Fig. R8**. The results show that the MPL magnitude slightly decreases immediately after the

exposure to air, and it is further reduced to a level of $\sim 25\%$ after a long-time (24 hours) exposure of RMCs in ambient condition.

Fig. R8. The $MPL(B)$ curves measured from RMC-1.4 μm in vacuum and in the air.

Accordingly, **Fig. R8** is added as Supplementary **Fig. 6** in Supplementary Information. The above discussion has also been mentioned in the revised main text (page 4) and in Supplementary Information.

6、*The authors should examine the influence of excitation power on MPL, because the process of triplet fusion is known to be power-dependent.*

Our Response: According to Reviewer #1's comments, we have measured the power-dependent ($10\text{-}1000\text{ mW/mm}^2$) MPL responses in RMCs, as shown in **Fig. R9**. It is obvious that the MPL magnitude shows a very slight decrease only at high excitation power, which indicates the bimolecular process of non-geminate TF should not be involved in the giant MPL of RMCs. As suggested by Reviewer #2, the MPL effect is related to SF and geminate TF which are both monomolecular processes, and non-geminate TF would contribute to an opposite MPL effect. Therefore, we have re-considered our interpretations on giant and size-dependent MPL in the revised version, which are now attributed to the spin state conversion of geminate SF and TF between singlets and triplet-triplet pairs in the excited-state processes.

Accordingly, **Fig. R9** is added as Supplementary **Fig. 17** in Supplementary

Information. The above discussion has also been mentioned in the revised main text (page 6 and page 8) as well as in Supplementary Information.

Fig. R9. MPL magnitude at 10 mT as a function of excitation power density used in our experiments for RMC-1.4 μm .

7. The concept of magneto-optical device shown in Figure 4 is interesting. However, I am a bit confused by the device configuration. Why the rubrene crystal needs to be so long? The active area should be between the two optical fibers, so the crystal on two sides would introduce additional loss due to optical waveguiding.

Our Response: According to Reviewer #1's comments, we have constructed an on-chip optical magnetometer of RMCs embedded in the integrated photonic circuits of PMMA waveguides. As shown in **Fig. R10**, RMCs with a length of 50 μm are fabricated on the pre-patterned PMMA waveguides, and the light signals are coupled from/to the network of optical waveguides on the photonic chip. The on-chip optical magnetometer is used to detect ultralow magnetic fields (at the level of $\sim\mu\text{T}$) at various frequencies (Hz to MHz) generated by a Helmholtz coil.

Fig. R10. Optical microscopy image of the on-chip RMC magnetometer integrated with PMMA waveguide and the zoom-in image on the RMC magnetometer. Scale bar is 50 μm .

Accordingly, **Fig. 4a** has been replaced with **Fig. R10** in the revised manuscript. The discussion and details have also been added in the revised main text (page 9).

8. *The >40% MPL is a remarkable value as far as I know. I recommend the authors to provide a summary of MPL values in previous studies for comparison. It would more clearly show the significance of this work in terms of very large magnetic-optical effects.*

Our Response: According to Reviewer #1's comments, we have summarized the MPL values in previous studies for comparison, as shown in **Fig. R11**. It is obvious that our results represent an important step forward to giant magneto-optical effects at ultralow fields in organic molecular materials.

Accordingly, **Fig. R11** is added as **Supplementary Fig. 9** in Supplementary Information. The above discussion have also been mentioned in the revised main text (page 4) and in Supplementary Information.

Fig. R11. Summary of the MPL effects observed in organic molecular materials from previous studies (references provided in Supplementary Materials).

Response to Reviewer #2

This manuscript reports the giant magneto-photoluminescence in rubrene microcrystals at relatively low magnetic fields and its application to a novel magneto-optical device. Such an intriguing magneto-photoluminescence has not been reported for organic crystalline materials and is of significance to the field of spin science as well as broad field of photo-functional materials. However, it seems that there are some critical flaws in data analysis, interpretation, and conclusions. The authors claim that the giant MPL is due to confinement of the triplet excitons in the microcrystal, but this conclusion should be drawn from more careful analysis of the experimental results. I think that the manuscript is suitable to publication in Nature Communications if the authors address ALL the comments below and revise the manuscript. Additional experiments may be needed for supporting their conclusions. I am eager to re-review the revised version of the manuscript.

Our Response: We are very grateful to Reviewer #2 for the overall positive evaluation as well as his/her time and efforts in examining our work. In the revised version, we have re-considered our interpretations carefully according to Reviewer #2's comments, and now we attribute the giant MPL to the geminate SF and TF between singlets and triplet-triplet pairs. The size of RMCs is related to the competition between SF and radiative decay of singlet excitons, and also to the competition between geminate TF and dissociation of triplet-triplet pairs into triplet excitons. The detailed responses to Reviewer #2's comments are provided as below, and the main text and Supplementary Information have been revised accordingly.

(1) It is well-known that the MPL due to the SF and TF in the crystalline materials is very sensitive to the orientation of the magnetic field relative to the crystal axis (see for example, R. C. Johnson and R. E. Merrifield, Phys. Rev. B 1 (1970) 896. or more recently, K. Ishikawa, T. Yago and M. Wakasa, J. Phys. Chem. C 122 (2018) 22264). I was pretty surprised that the authors do not state clearly about the orientation of the

magnetic field relative to the crystal axis. For the RMC, I can only see “in-plane” in line 102, but XRD and SAED of the RMC indicates “ordered molecular packing in both out-of-plane and in-plane directions”. In plane angle relative to the crystal axis is also important. Then the MPL must be compared with the MPL for the bulk single crystal at exactly the same relative angle between the crystal axis and the magnetic field. I suspect that the large MPL is partially due to critical matching of the field direction to the resonant field.

Our Response: We agree with Reviewer #2 that MPL in crystals should be sensitive to the direction of external field, especially because the SF and TF are related to the intermolecular distance in ordered molecular packing. As pointed out by Reviewer #2, the anisotropy of MPL has been observed previously in anthracene crystals [Ref. 42, Johnson, R. C. *et al. Phys. Rev. B* **1**, 896-902 (1970)], in DPH crystals [Ref. 44, Ishikawa, K. *et al. J. Phys. Chem. C* **122**, 22264-22272 (2018)], or more relevant to rubrene, in tetracene crystals [Ref. 43, Merrifield, R. E. *et al. Chem. Phys. Lett.* **3**, 386-388 (1969)]. In rubrene crystals, to the best of our knowledge, the anisotropic MPL has not yet been reported. Compared with the tetracene crystal structure, the molecular packing of rubrene is less anisotropic especially in its triclinic phase, as shown in Fig. R12. In general, the reduced structural anisotropy (in terms of the intermolecular distance along crystal axis) should lead to the reduced anisotropy in the photophysics properties of rubrene crystals. For instance, the PL anisotropy under linearly polarized excitation is reported to be 1: 0.17 in tetracene crystals [Tavazzi, S. *et al. J. Chem. Phys.* **128**, 154709 (2008)], and it is 1: 0.33 in rubrene crystals [Tavazzi, S. *et al. Phys. Rev. B* **75**, 245416 (2007)]. Therefore, the MPL anisotropy in rubrene crystals is expected to exist and be probably weaker than that in tetracene crystals.

Fig. R12. Molecular packing of (a) tetracene crystal and (b) rubrene crystal.

In the revised version, we have supplemented the MPL measurements by changing the in-plane angle θ of external field relative to the long axis of rubrene crystals for both RMCs (**Fig. R13**) and bulk samples (**Fig. R14**). The long axis is along the (111) direction of triclinic phase rubrene crystal. In RMCs, the anisotropy of MPL is observed but relatively weak compared to the overall large MPL magnitude ($\sim 40\%$). The MPL maximum appears at 40° and 130° , and the change of MPL magnitudes up on the direction of external field is $\sim 3\%$. In comparison, the MPL of rubrene bulk crystals is more anisotropic; i.e., the maximum at 40° and 130° is $\sim 17\%$ while the minimum is $\sim 12\%$ at 10 mT. It is relatively comparable to that observed for MPL in tetracene crystals, which varies from 1% to 3% at a low field of 15 mT, and from 10% to 35% at a high field of 500 mT. [Bouchriha, H. *et al. J. Phys. France* **39**, 257-271 (1978)]. Note that the MPL responses of RMCs and rubrene crystals follow the same angle dependence, thus the size effects of MPL observed in this work is not related to the MPL anisotropy. The critical matching of the field direction to the resonant field partially contributes to the giant MPL in RMCs, but its contribution is quite limited ($3\sim 5\%$) compared with the large MPL magnitude of $\sim 40\%$.

Fig. R13. (a) Fluorescence image of single RMC-1.4 μm . Scale bar is 5 μm . The definition of rotation angel θ according to the long axis along (111) of RMC. (b) Corresponding MPL magnitude obtained at 10 mT as a function of the rotation angle of magnetic field.

Fig. R14. (a) Fluorescence image of single triclinic rubrene crystal. Scale bar is 20 μm . The definition of rotation angel θ according to the long axis along (111) of the crystal. (b) Corresponding MPL magnitude obtained at 10 mT as a function of the rotation angle of magnetic field.

Accordingly, **Fig. R13** and **Fig. R14** are added as **Supplementary Fig. 23** and **Supplementary Fig. 22** in **Supplementary Materials**. The above discussion and citations (**Ref. 42** to **Ref. 44**) have also been mentioned in the revised main text (page 6) and added in **Supplementary Materials**.

(2) This is somewhat relevant to the comment (1), but comparison with the polycrystalline film is also very rough. Using the Scherrer's equation, one can estimate the grain size in the film from the XRD result. The grain size may be micron order, which

should be suitable to the “confinement”. The spatially resolved MPL shown in Figure S5 also indicates the existence of micron-order grains.

Our Response: According to Reviewer #2’s comments, we have estimated the grain size using the Scherrer’s equation for the XRD result, as shown in **Fig. R15**. The grain size is about 50-70 nm for the polycrystalline film, and we have also measured the size of crystal grains according to the SEM image.

In the revised version, we attribute the giant MPL to the spin conversion of geminate SF and TF between singlets and triplet-triplet pairs in RMCs. In the polycrystalline film, the unavoidable defects introduce nonradiative decay routes for both singlets and triplet-triplet pairs, thus it would suppress the geminate SF and TF for MPL. This feature is confirmed by the prolonged exciton lifetime in RMCs compared to that in polycrystalline film according to the TA results (Supplementary **Fig. 25**).

Fig. R15. (a) Grain size of polycrystalline rubrene calculated from XRD. (b) SEM image of polycrystalline rubrene. Scale bar is 1 μm .

Accordingly, **Fig. R15** is added as Supplementary **Fig. 3** in Supplementary Materials. The above discussion has also been mentioned in the revised main text (page 4 and page 8) and added in Supplementary Materials.

(3) I felt the term “bimolecular” is confusingly used in the whole manuscript. Both the SF and TF do require two adjacent chromophore molecules, but in terms of reaction

kinetics in the photoexcited state, SF is definitely monomolecular. TF is also monomolecular if the T-T pair is generated only by SF (geminate T-T pair), but it can be bimolecular if the non-geminate T-T pair is formed followed by long-distance triplet diffusion. The non-geminate TF is thus highly sensitive to the excitation photon density, which is not described in the manuscript. I assume that uniform excitation of the crystals is difficult because of the strong absorption (steep decay of the excitation density from the surface), but comparison of the results should be done carefully maintaining the excitation density in the crystals/films. Time-resolved measurement at pulse excitation condition (Fig 3b) indicates the existence of long delayed fluorescence, which is likely due to the bimolecular TF. Since the geminate SF and TF is monomolecular, laser power dependence of the MPL should be constant in the low power limit, where the bimolecular TF becomes negligible. This would give a clear evidence that T-T pair dynamics is modulated by confinement of the excitons in the RMC. If only the bimolecular TF is responsible to the “confinement effect”, the authors should change the conclusion. Namely, the size dependence of the MPL is simply due to difference in the excitation density. It should also be noted that SF and non-geminate TF give the opposite MPL.

Our Response: Reviewer #2 is correct that SF and TF need two molecules to take place but geminate SF and TF should be monomolecular in terms of kinetics, and the word “bimolecular” was confusingly used in our original manuscript. We have corrected the related parts and clarified the descriptions of SF and TF in the revised version.

Besides, we have supplemented the power dependence of PL intensity and MPL magnitude in polycrystalline film (**Fig. R16**) and RMCs (**Fig. R17**). As shown in **Fig. R16**, the PL emission from polycrystalline film is dominated by the monomolecular process in the range of excitation power from 10 to 1000 mW/cm², as its power dependence is perfectly fitted with a slope of 1. It confirms the absence of non-geminate TF process due to the relatively inefficient pre-occurred SF process in polycrystalline

films. Note that the MPL magnitude remains almost unchanged with the increase of excitation power. In comparison, the PL emission of RMCs is also monomolecular in a wide range of excitation power, and its dependence slightly deviates from the linear relation only under high excitation density, as shown in **Fig. R17**. Meanwhile, the MPL magnitude slightly decreases under high excitation density, which could be attributed to the opposite MPL from non-geminate TF process. It should be mentioned that in our work all typical MPL measurements were carried out at the excitation power density of $\sim 100 \text{ mW/mm}^2$. In this case, the non-geminate TF and its contribution to MPL should be negligible in both polycrystalline films and RMCs, and the MPL magnitude in RMCs is significantly larger than that in polycrystalline films due to the suppression of nonradiative decay in RMCs. Thus, it can be concluded that the geminate SF and TF process should be responsible for the size-dependent MPL performance in RMCs.

Fig. R16. (a) PL intensity as a function of input excitation power density measured from polycrystalline rubrene film. (b) MPL magnitude at 10 mT as a function of excitation power density measured from polycrystalline rubrene film.

Fig. R17. (a) PL intensity as a function of input excitation power density measured from RMC-1.4 μm . (b) MPL magnitude at 10 mT as a function of excitation power density measured from RMC-1.4 μm .

In **Fig. 3b**, the PL decay was divided into two regimes, prompt PL and delayed PL, to illustrate the competition between SF process and radiative decay of singlet excitons, and the competition between geminate TF and dissociation of triplet-triplet pairs into free triplets, respectively. It should be mentioned that the regime of delayed PL is enlarged to show its dynamics more clearly, and the intensity of delayed PL is in fact much lower than that of prompt PL. This is in consistence with the negligible contribution from non-geminate TF to the giant MPL in RMCs. Therefore, the size effect of MPL is related to the photophysical processes in RMCs and can be interpreted as follow. “(i) For RMC-0.1 μm , the direct radiative recombination of S_1 is more efficient than the SF process into (T_1-T_1) for spin state conversion; (ii) For RMC-1.4 μm , SF process is facilitated and allows for the reversible conversion between S_1 and (T_1-T_1) through geminate SF and TF processes; (iii) For RMC-2.0 μm , the dissociation of (T_1-T_1) into T_1 is more pronounced and diminishes the spin state conversion, in addition, T_1 participates in non-geminate TF that brings an opposite MPL effect.”

Accordingly, **Fig. R16** and **Fig. R17** are added in Supplementary **Fig. 18** and Supplementary **Fig. 17** in Supplementary Materials. The above discussion has also been mentioned in the revised main text (page 6 and page 8) and added in supplementary materials.

(4) With regard to the temperature dependence of MPL starting from line 204, I totally agree with the conclusion that monomolecular PL (this should be called prompt PL or monomer PL since SF is also monomolecular as stated above) becomes dominant over SF at lower temperatures, which results in decrease in MPL. However, analysis of the PL in comparison with that in the polymer matrix to obtain “the number of triplet excitons in RMC” does not make any sense to me. The authors should explain more in detail regarding the analysis method in the supporting information. Figure 3e lacks the numbers for the left axis, so that comparison with MPL is difficult. At least, the zero line is needed. Anyway, this part does not support the conclusion regarding “confinement of the T-T pair” at all. Thermally-activated SF in the rubrene crystal is a well-known fact (see for example, L. Ma, K. Zhang, C. Kloc, H. Sun, C. Soci, M. E. Michel-Beyerle, and G. G. Gurzadyan, *Phys. Rev. B* 87 (2013) 201203).

Our Response: We agree with Reviewer #2 that the temperature dependence should be attributed to thermally-activated SF in the rubrene crystals. [Ref. 57, Ma, L. *et al. Phys. Rev. B* 87, 201203 (2013)]. According to Reviewer #2’s comments, we have revised the part related to the temperature dependent results of RMCs in **Fig. 3e**. The competition between SF process and direct radiative recombination of singlet excitons is profiled by the averaging lifetime of prompt PL, τ_{prompt} . By measuring the rubrene monomer dispersed in PMMA (a solid-state solution for low-temperature measurement), we can obtain the PL lifetime solely from direct radiative recombination of singlet excitons, τ_{monomer} . The difference between two lifetime parameters is thus in proportional to the portion of singlet fission in the excited state processes of singlet excitons. As the obtained τ_{monomer} shows a very weak dependence on temperature (**Fig. R18**), it is thus reasonable to use τ_{prompt} to describe the contribution from SF process. The shorter τ_{prompt} is, the more singlets undergo SF. Therefore, we’ve plotted the temperature dependence of τ_{prompt} and MPL magnitude together in **Fig. R19**. It shows that the two curves follow a similar trend, which indicates the MPL effect is semi-quantitatively associated with the geminate SF and TF processes in RMCs. Note that the same analysis method is also

applied to the size-dependent PL lifetimes and MPL magnitudes in RMCs.

Fig. R18. Transient PL decay curves measured at different temperatures from rubrene monomer dispersed in PMMA.

Fig. R19. Averaging lifetime of prompt PL (black) and MPL magnitude (red, at 10 mT) as a function of temperature for RMC-1.4 μm .

Accordingly, raw data of temperature-dependent PL dynamics of monomer rubrene in **Fig. R18** were included as Supplementary **Fig. 30** in Supplementary Materials. We have also replaced **Fig. 3e** with **Fig. R19** and added the above discussion and citation (**Ref. 57**) in the main text (page 8).

(5) In total, I suspect that the confinement of T-T pair in the RMC does not contribute to the large MPL. One more evidence for this is clearly shown in Figure 4b. The large MPL at the low magnetic field rises in nanosecond timescale. Within such a short time, the triplet exciton generated by SF can diffuse only a few tens of nanometers, so that the micrometer-order crystal size does not affect the T-T pair dynamics. The

bimolecular TF, which should be observed at later times, could be affected by the crystal size, but this effect should be carefully separated from the simple density effect as explained in the comment 3.

Our Response: We agree with Reviewer #2 that the confinement of triplet excitons should not contribute to the giant MPL observed in RMCs. Hence, we contribute the size effect and the giant MPL to the spin conversion of geminate SF and TF processes between singlets and triplet-triplet pairs, since the size of crystals has been proven to influence the photophysical processes including SF and TF in rubrene [**Ref. 54**, Gieseck, B. *et al. Phys. Rev. B* **90**, 205305 (2014)].

Our interpretation in the revised version is further illustrated in **Fig. R20**. It can be universally used in analyzing the experimental data we obtained in this work. (i) In polycrystalline film, the defects introduce non-radiative decay routes for both singlets and triplet-triplet pairs, which is unfavorable for the spin conversion of geminate SF and TF and diminish the MPL effect (1%). (ii) In RMCs, the competition of physical processes in singlets (direct recombination and SF) and in triplet-triplet pairs (geminate TF and dissociation into triplets) determines the MPL magnitude (up to 40%), as shown in **Fig. R21**. The dissociation into triplets and consequently the non-geminate TF are pronounced in rubrene bulk crystals, and thus lead to a reduced MPL magnitude (<20%). (iii) The SF process is thermally activated, so the MPL effect is significantly reduced at low temperature. The time-resolved measurement shows MPL takes place within a few nanoseconds that should be attributed to geminate SF and TF, and the MPL effect is opposite at a longer time delay that is assigned to non-geminate TF of triplets.

Fig. R20. The emission and spin conversion processes in rubrene. The promoted geminate SF and TF induce the giant MPL as the monomer PL and dissociation of triplet-triplet pairs are suppressed in RMCs with suitable thickness. Note that the non-radiative process is also suppressed in the high quality rubrene crystals.

Fig. R21. The lifetime of prompt PL (τ_{prompt} , blue) and the power function factor of delayed PL (m_{delayed} , red) as a function of the thickness of RMCs. Illustration for the size effect of SF and TF processes in RMCs is shown in the inset.

Accordingly, we have included the original **Fig. 4b** in the revised **Fig. 3** and replaced **Fig. 3c** with **Fig. R21** in the manuscript. **Fig. R20** has also replaced Supplementary **Fig. 1** in Supplementary Materials. The above discussion and citation (**Ref. 54**) are also added in the main text (page 8) as well as in Supplementary Materials.

Response to Reviewer #3

In the present manuscript, the authors prepared rubrene microcrystals using a capillary-bridge assembly method. Magnetic field effects on luminescence from the rubrene microcrystals were observed with a home-built optical microscope equipped with a magneto-optical cryostat. The decrease of fluorescence by 40 % was observed at 100 mT. Fluorescence intensity was modulated by oscillating external magnetic field using Helmholtz coil.

Although physical phenomena observed in the present manuscript may be interesting, the general readers of nature communications will not understand or be interested in this topic. I do not recommend publication of the present manuscript in Nature Communications for the following reasons.

Our Response: We appreciate Reviewer #3 for his/her time and efforts in examining our work, as well as recognizing that the physical phenomena are new and interesting. The comments are indeed helpful for us to substantially improve the manuscript.

The main achievements presented in the revised version of our work include: (i) The giant MPL over 40% at a low field of 10 mT is observed in rubrene microcrystals (RMCs), which is far above previously reported values. (ii) The unique size dependence of giant MPL encourages us to develop a capillary-bridge assembly method in preparing high-crystallinity arrays of size-tunable RMCs. (iii) RMCs are patterned on the network of optical waveguides to construct an on-chip magnetometer for optically detecting magnetic fields at a broad range of frequencies.

Nature Communications is a multidisciplinary journal dedicated to publishing high-quality research in different areas of basic and engineering sciences and have a wide range of readership. In our humble opinion, the studies on new and interesting physical phenomena can lead to the development of research fields, which is attractive

to the general readership in physical sciences. This is also true for the area of organic opto-spintronics concerning the observation of magnetic field effects on excited state process [e.g., Wang, Z. W. *et al. Nat. Chem.* **13**, 559-567 (2021); Izawa, S. and Hiramoto, M. *Nat. Photonics* **15**, 895-900 (2021); Dou, Y. X. *et al. Nat. Commun.* **12**, 3485 (2021); Lafalce, E. *et al. Nat. Commun.* **13**, 483 (2022)], in which the spin conversion of excited states is proven to be very important for the development of fascinating opto-electronic materials/devices in recent years. We understand that Reviewer #3 is critical on the original manuscript considering the high criteria and the broad readership of *Nature Communications*, which is exactly we are re-considering and trying our best to achieve by supplementing new results and strengthening our work.

In the revised manuscript, we have demonstrated the use of interesting giant MPL phenomenon in the construction of on-chip optical magnetometers, based on the precise control of RMCs in our capillary-bridge assembly method. To fully understand the size effects on MPL phenomenon, we have supplemented a series of additional experiments (e.g., anisotropic MPL, power dependent MPL). In addition, we have re-organized the manuscript to clarify the significance and novelty of our work in the areas of organic opto-spintronics and integrated photonics. Our work is now not only featured by the observation of giant MPL and its underlying mechanism, but also include the advances in the controllable preparation of giant MPL materials as well as the on-chip fabrication of optical magnetometers using solution assembly technique.

We hope our efforts make sense to the reviewers. We believe the overall quality of our work has been substantially improved, and it represents an important step forward in both the theory and the application of organic magneto-optical effects. Detailed responses to Reviewer #3 are as below.

1. The authors claimed that the present material can be a prototype of emerging opto-spintronic devices. However, I can't imagine exactly how this material would be useful for device development. In introduction, the authors also mention about magneto-

photoluminescence with several literatures. In these literatures, however, it is not explained how magneto-photoluminescence improves existing devices or contributes to develop new functional devices. In addition to that, I do not think that luminescence from the rubrene crystal is coherent, which is important in development of quantum optical devices.

Our Response: Magneto-photoluminescence (MPL), a typical example of magnetic field effects (MFEs), has been widely used to study spin-related process for device applications in spintronics and opto-spintronics. According to Reviewer #3's comments, we have supplemented a few cases of MPL/MFEs for device development. For improving the existing devices, MPL/MFEs are useful to profile the excited state processes, and improve the performance of light emitting diodes [Wang, M. S. *et al. Nat. Commun.* **10**, 1614 (2019)] and photovoltaic devices [Wang, J. Y. *et al. Nat. Commun.* **10**, 129 (2019)]. For developing new functional devices, MPL/MFEs have been adopted in the design of magneto-optical isolator [Yuan, S. X. *et al. Nat. Commun.* **12**, 5570 (2021)] and magnetometry [Kim, D. *et al. Nat. Electron.* **2**, 284-289 (2019)].

In the revised version, we focus on the use of giant MPL in RMCs for the on-chip optical magnetometer, which is an integrated optical sensor (classic devices) and does not require the coherence of light signal. The integrated on-chip optical magnetometer of RMCs is embedded in the integrated photonic circuits of PMMA waveguides to detect ultralow magnetic field ($\sim\mu\text{T}$) at a broad range of frequencies (Hz to MHz). Besides, the SF/TF process leads to the observation of exciton quantum beats [Wang, R. *et al. Nat. Commun.* **6**, 8602 (2015)], in which the spin-corelated triplet-triplet pair is possibly useful for the development of quantum optical devices [Gorgon, S. *et al. Nature* **620**, 538-544 (2023)]. This is not our goal here, but definitely could be very exciting attempts in our future works.

Accordingly, we have revised the Introduction part to emphasize the advantages of our assembly techniques and to emphasize the potential applications of giant

MPL/MFEs. Some relevant citations are also added in the main text (**Ref. 12** and **Ref. 16**) to clarify the novelty and significance of our work in the field of opto-spintronics devices.

2. The authors claimed that the observed large magnetic field effects were due to the confinement of triplet excitons in micro-space. In our laboratory, we also measured magnetic field effects on fluorescence from rubrene crystals with a variation of crystal phase. The figure below shows an example of our data. Although the crystals are visible, we can observe the 25 % decrease at 50 mT. Therefore I can not believe that the magnetic field effects in the present manuscript is purely caused by the confinement of excitons. The authors changed the size of the materials by applying pressure. The pressure also changes the packing of crystals. It is not clear whether magneto-photoluminescence is affected by packing change or size change of the crystals.

Our Response: It is correct that MPL could be affected by either packing change or size change of the rubrene crystals. In our work, the RMCs fabricated by capillary-bridge assembly method (**Fig. R22a**) are all in triclinic phase with excellent crystallinity, and the molecular packing is expected to remain the same. To explain our assembly method more clearly and intuitively, the assembly process of RMCs was observed and characterized by fluorescence microscopy, as shown in **Fig. R22b**. The liquid thin film is formed and confined between the target substrate and the micropillar-structure photoresist template under a static pressure. With the evaporation of solvents, the receding of triphase contact line happens at the gaps between two adjacent columns of micropillars driven by the Laplace force. Then the liquid is separated into individual rows and the isolated capillary bridges are anchored onto each individual micropillar. The fluorescence microscopy image of the capillary bridges exhibits uniform fluorescence, indicating the fluid field characteristic and mass transport direction are all the same for each individual capillary bridge. When the rubrene solution becomes supersaturated, the nucleation and growth of rubrene crystals happen in the isolated

capillary bridges.

Fig. R22. (a) Schematic diagram of the liquid bridge dewetting process. (b) The assembly process of RMCs under microscope. Scale bar is 5 μm .

Fig. R23. (a) Schematic diagram of RMCs nucleus growing in the confined space. (b) Corresponding environmental scanning electron microscopy (ESEM) images for the cross-sectional view of an individual micropillar with and without pressure. Scale bar is 1 μm .

As Reviewer #3 mentioned, a static pressure was applied in the capillary-bridge assembly system to control the size of confined space between the template micropillars and the underneath substrate. As shown in **Fig. R23a**, the pressure is applied against the surface tension from shape change of liquid film, and the crystals are grown within the liquid film on the substrate. It is verified by the ESEM images with cross-sectional view of an individual micropillar, as shown in **Fig. R23b**. Therefore, the net pressure

on the assembled crystals should be nearly zero, and should not influence the packing of rubrene molecules [Ref. 50, Gao, H. F. *et al. Nat. Commun.* **10**, 3912 (2019)]. Especially, the in-situ SEM image of completely crystallized structures shows a gap between the assembled crystal and the micropillar, which supports that the nucleation/growth processes are always in the liquid environment and unaffected by external pressure.

To further address Reviewer #3's concerns, we have supplemented the AFM images and corresponding TEM and SAED results on the RMCs with different thicknesses (0.2, 0.5, 1.9 μm , prepared under various pressures), as shown in Fig. R24. It is clear that the molecular packing remains all the same in triclinic phase in these samples of RMCs. In addition, the GIWAXS measurements were also carried out in these RMCs and indicating the same crystal orientation of them.

Fig. R24. (a) AFM images of three adjacent RMCs with the width of *ca.* 2 μm but different heights varying from 200 nm to 1900 nm. (b) TEM image and SAED pattern of corresponding RMCs with different heights. Scale bar is 1 μm . (c) GIWAXS pattern of corresponding RMCs.

To validate the size effects of MPL in rubrene, we have compared the MPL results of RMCs and bulk crystals (the molecular packing is the same) using the same setup. The statistics of MPL performance in a variety of rubrene samples is also provided in **Fig. R25**. It is obvious that the MPL magnitude of RMCs can be greatly enhanced compared with that of bulk crystals. The size effect is attributed to the size-dependent photophysical processes in RMCs [Ref. 54, Gieseeking, B. *et al. Phys. Rev. B* **90**, 205305 (2014)], i.e., the competition between direct recombination and SF of singlets, as well as the competition between geminate TF and dissociation into triplets of triplet-triplet pairs. In addition, we have carried out a series of control experiments to confirm the size effects, such as power-dependent, angle-dependent, temperature-dependent and time-resolved measurements on MPL effects (provided as Supplementary Fig. 17-18, Supplementary Fig. 22-23, Supplementary Fig. 32 and Fig. 3d in the revised version).

Fig. R25. (a) MPL(B) curves measured from RMC-1.4 μm . (b) MPL(B) curves measured from triclinic rubrene bulk crystal. (c) Statistics of MPL magnitudes distribution measured from bulk crystals and RMCs.

We would like to mention that our group has the expertise in measuring various magnetic field effects [*Nat. Phys.* **11**, 427-434 (2015); *Angew. Chem. Int. Ed.* **62**, e2023090 (2023); *Adv. Funct. Mater.* **33**, 2211059 (2023); *Phys. Rev. Appl.* **21**, 014039 (2024).]. We are aware that the artifacts in MPL measurement are tricky, and even the reported MPL values for the same material are sometimes varied from sample to sample. In our lab, the MPL setup has been carefully examined and calibrated with many previously reported materials, and we are sure that the MPL results presented in this work are highly reproducible and reliable.

Accordingly, **Fig. R23** and **Fig. R24** are added as Supplementary **Fig. 11** and Supplementary **Fig. 12**, and **Fig. R22** and **Fig. R25** has replaced Supplementary **Fig. 2** and Supplementary **Fig. 20** correspondingly in Supplementary Materials. The above discussion has also been mentioned in the revised main text (page 4 to page 5) and added in Supplementary Materials.

3. The modulation of the fluorescence by the oscillating magnetic field has been used to study low magnetic field effects on radical pairs (C. T. Rodgers, Pure Appl Chem. 2009, 81, 19-43). Therefore, the technique in the manuscript is not new. I can not imagine how it works in organic devices. What means a magneto-optical integrated device?

Our Response: In this work, we present the giant MPL in RMCs prepared by the capillary-bridge assembly method. The on-chip magneto-optical devices are fabricated to demonstrate the potentials of giant MPL in the integrated photonics, which benefits from the facile fabrication of RMCs that is compatible with integrated photonic circuits. The oscillating magnetic field generated by Helmholtz coil is a widely used technique [added as **Ref. 58**, Rodgers, C. T. *Pure Appl Chem.* **81**, 19-43 (2009), and it was used for studying photogenerated radical pairs for magnetic compass], and it is adopted here to generate ultralow fields at various frequencies for device application based on RMCs.

To better demonstrate the device application of giant MPL in RMCs, we have re-considered the design of our proof-of-concept devices and constructed an on-chip optical magnetometer integrated with the photonic circuits of PMMA waveguides. The on-chip optical magnetometer [Ref. 17, Budker, D. and Romalis, M. *Nat. Phys.* **3**, 227-234 (2007)] is used to detect ultralow magnetic fields (at the level of $\sim\mu\text{T}$) at a broad range of frequencies (Hz to MHz), and its parameters are comparable to those reported in the state-of-art integrated optical magnetometers. [Ref. 18, Gotardo, F. *et al. Opt. Express* **31**, 37663-37672 (2023)]

As shown in **Fig. R26a**, RMCs with a length of 50 μm are fabricated on the pre-patterned PMMA waveguides, and the light signals are coupled from/to the network of optical waveguides on the photonic chip. The efficient near-field coupling between the RMC and the PMMA waveguide ensures the detection of low magnetic field at different frequencies generated by a Helmholtz coil. In the RMC-based on-chip magnetometer, the light passing RMC is considered as the signal while the light through the bypass waveguide is used as the reference for zero-field intensity. In principle, the nanosecond response of MPL in RMCs offers possibilities for the magnetic field detection up to $\sim\text{GHz}$ frequency. As shown in **Fig. R26b**, a cosine-modulated light signal at 2 kHz is outcoupled from the PMMA network when the alternative field between -500 μT and +500 μT is applied at 1 kHz. Meanwhile, the reference is constant under external field and used to determine the noise level of optical magnetometer. As shown in **Fig. R26c**, the external field is detected without any obvious attenuation of signal amplitude in a broad range of frequencies ramping up to 0.1 MHz, which is actually limited by the highest frequency in operating the Helmholtz coil. Moreover, the RMC magnetometer can detect a low field down to $\sim\mu\text{T}$ level benefiting from the giant MPL effect, and the waveshapes of alternative fields (e.g., trapezoid, triangle) can even be accurately recognized. The signal to noise ratio (SNR), $\text{SNR}=10\times\lg(\text{MPL}_{\text{signal}}/\text{NOISE}_{\text{reference}})$, is adopted to profile the sensitivity of on-chip optical magnetometer [Mousavi, M. *et al. Small* **20**, 2304591 (2023)], as shown in **Fig. R26d**. The SNR values follow a power-

law function of the field strength in a broad range of frequencies from Hz to MHz, and the intercept represents the limit of detection below $10 \mu\text{T}$. The above results demonstrate a prototype of optical integrated magnetometer for detecting ultralow fields at a wide range of frequencies, taking advantages of unique fabrication technique and magneto-photophysical property from organic semiconductors.

Fig. R26. (a) Optical microscopy image of the on-chip RMC magnetometer and the zoom-in image on the on-chip RMC magnetometer. Scale bar is $50 \mu\text{m}$. (b) Response of RMC magnetometer in signal and reference under applied alternative field. (c) The detected signal response from the on-chip RMC magnetometer at different frequency, intensity and waveshape of alternative field. (d) SNR of RMC magnetometer under different frequency of 1 Hz, 1 kHz and 0.1 MHz applied magnetic field.

Accordingly, we have replaced **Fig. 4** with **Fig. R26**, and the above discussion and citations (**Ref. 17, Ref.18, Ref. 58**) has been included in the revised main text (page 9).

4. In page 9, Discussion should be Summary.

Our Response: According to Reviewer #3's comments, we have changed the section title to Summary in the main text.

Reviewer #1 (Remarks to the Author):

My questions have been well addressed. I recommend the publication of the work in the Nature Communications.

Reviewer #2 (Remarks to the Author):

This manuscript reports the giant magneto-photoluminescence in rubrene microcrystals at relatively low magnetic fields and its application to a novel magneto-optical device. Such an intriguing magneto-photoluminescence has not been reported for organic crystalline materials and is of significance to the field of spin science as well as broad field of photo-functional materials. In the resubmitted version of the manuscript, the authors have added a volume of experimental data and analysis of them. The interpretation of the experimental data seems very reasonable, and the newly developed magneto-optical device is fascinating, so that I strongly recommend this manuscript for publication in Nature Communications. However, I still do not understand the intrinsic mechanism of the crystal size effect. I agree that the magnitude of MPL is determined by the balance of the rates for fission, T-T pair fusion and dissociation, but how these processes are controlled by the crystal thickness is still unclear to me. It would be better if the authors could comment on this issue from their results or published data.

Reviewer #3 (Remarks to the Author):

Thank you for replying about my opinions and for revising your paper according to my opinions. I feel that my opinions are fully reflected in the revised manuscript. The manuscript can be accepted without further revision.

Response to Reviewers' comments

Response to Reviewer #1

My questions have been well addressed. I recommend the publication of the work in the Nature Communications.

Our Response: We appreciate Reviewer #1 again for his/her positive evaluation.

Response to Reviewer #2

This manuscript reports the giant magneto-photoluminescence in rubrene microcrystals at relatively low magnetic fields and its application to a novel magneto-optical device. Such an intriguing magneto-photoluminescence has not been reported for organic crystalline materials and is of significance to the field of spin science as well as broad field of photo-functional materials.

In the resubmitted version of the manuscript, the authors have added a volume of experimental data and analysis of them. The interpretation of the experimental data seems very reasonable, and the newly developed magneto-optical device is fascinating, so that I strongly recommend this manuscript for publication in Nature Communications.

Our Response: We appreciate Reviewer #2 again for his/her positive evaluation.

However, I still do not understand the intrinsic mechanism of the crystal size effect. I agree that the magnitude of MPL is determined by the balance of the rates for fission, T-T pair fusion and dissociation, but how these processes are controlled by the crystal thickness is still unclear to me. It would be better if the authors could comment on this issue from their results or published data.

Our Response: We agree with Reviewer #2 that how SF/TF processes are controlled

by the size of microcrystals remains an open question. A plausible answer that we can provide is related to the cooperative singlet and triplet exciton transport [added as Ref. 57, Wan, Y. *et al. Nat. Chem.* **7**, 785-792 (2015)]. It could occur on the same scale of RMC thickness especially when the penetration depth of excitation light is considered. In this case, the SF/TF processes involving reversible singlet-triplet conversion are expected to depend on the thickness of RMCs [Ref. 54, Gieseeking, B. *et al. Phys. Rev. B* **90**, 205305 (2014)]. It would be worthwhile to conduct further studies to understand the detailed mechanism for size-dependent photophysical processes in RMCs.

Accordingly, we have added the following discussion and Ref. 57 on page 6 in the main text: “The size-dependent photophysical properties are probably due to the cooperative singlet and triplet exciton transport^[57], which could occur on the same scale of RMC thickness especially considering the penetration depth of excitation light.”

Response to Reviewer #3

Thank you for replying about my opinions and for revising your paper according to my opinions. I feel that my opinions are fully reflected in the revised manuscript. The manuscript can be accepted without further revision.

Our Response: We are glad to know our revisions make sense to Reviewer #3, and the manuscript is now recommended to be accepted without further revision.